# β-Tubulin carboxy-terminal tails exhibit isotype-specific effects on microtubule dynamics in human gene-edited cells

Amelia L Parker[1,2,3], Wee Siang Teo[1,2,3], Elvis Pandzic[4], Juan Jesus Vicente[5], Joshua A McCarroll[1,2,3] , Linda Wordeman[5] , Maria Kavallaris[1,2,3]

**Microtubules are highly dynamic structures that play an integral role in fundamental cellular functions. Different α- and β-tubulin isotypes are thought to confer unique dynamic properties to microtubules. The tubulin isotypes have highly conserved structures, differing mainly in their carboxy-terminal (C-terminal) tail sequences. However, little is known about the importance of the C-terminal tail in regulating and coordinating microtubule dynamics. We developed syngeneic human cell models using gene editing to precisely modify the β-tubulin C-terminal tail region while preserving the endogenous microtubule network. Fluorescent microscopy of live cells, coupled with advanced image analysis, revealed that the β-tubulin C-terminal tails differentially coordinate the collective and individual dynamic behavior of microtubules by affecting microtubule growth rates and explorative microtubule assembly in an isotype-specific manner. Furthermore, βI- and βIII-tubulin C-terminal tails differentially regulate the sensitivity of microtubules to tubulin-binding agents and the microtubule depolymerizing protein mitotic centromere-associated kinesin. The sequence of the β-tubulin tail encodes regulatory information that instructs and coordinates microtubule dynamics, thereby fine-tuning microtubule dynamics to support cellular functions.**

## Introduction

The microtubule cytoskeleton is a dynamic intracellular structure composed of α- and β-tubulin heterodimers. The dynamic behavior of the microtubule cytoskeleton is critical in supporting cellular structure; in the transport of vesicles, proteins, and organelles; in enabling cell motility; and in ensuring correct segregation of the chromosomes during mitosis (Janke, 2014). In humans, microtubules are composed of mixtures of nine α-tubulin isotypes and nine β-tubulin isotypes, which each possess distinct tissue distributions (Verdier-Pinard et al, 2009; Luduena, 2013). For example, the βI-tubulin protein is ubiquitously expressed, whereas the βIII-tubulin protein is normally only expressed in neurons and testicular Sertoli cells (Kavallaris, 2010). The tubulin isotype composition forms a central component of the tubulin code, which together with posttranslational modifications and interactions with microtubule-associated proteins (MAPs), is hypothesized to form the regulatory mechanisms that specialize microtubule behavior (Gadadhar et al, 2017) but remains poorly defined.

The members of the tubulin protein family share a highly homologous structure, composed of a globular body formed from the N-terminal and intermediate domains, and a highly flexible and disordered acidic carboxy-terminal (C-terminal) tail region (Nogales, 2000). The C-terminal tail of the tubulin proteins extends outward from the wall of the microtubule, where it is a site for a wide range of posttranslational modifications and for interactions with proteins that regulate microtubule dynamics and other signaling effectors (Janke, 2014; Roll-Mecak, 2015). The C-terminal tails are the most divergent regions of the β-tubulin isotype sequence and serve to distinguish the tubulin isotypes from one another, making this region a prominent candidate in defining the isotype-specific function of the tubulin proteins.

Microtubule dynamics is partially regulated by the tubulin isotype composition. Studies in reduced cell-free systems using isolated tubulin and isotypically purified microtubules (Banerjee et al, 1994, 1997; Panda et al, 1994; Derry et al, 1997; Pamula et al, 2016; Vemu et al, 2017), and more recent in vivo studies (Honda et al, 2017) have determined that microtubules composed of different tubulin isotypes possess distinct dynamic behaviors. Of the β-tubulin isotypes, the βIII-tubulin isotype has been identified

[1]Children's Cancer Institute, Lowy Cancer Research Centre, University of New South Wales, Sydney, New South Wales, Australia    [2]Australian Centre for NanoMedicine and Australian Research Council Centre of Excellence for Convergent BioNano Science and Technology, University of New South Wales, Sydney, New South Wales, Australia [3]School of Women's and Children's Health, Faculty of Medicine, University of New South Wales, Sydney, New South Wales, Australia    [4]Biomedical Imaging Facility, Mark Wainwright Analytical Centre, Lowy Cancer Research Centre, University of New South Wales, Sydney, New South Wales, Australia    [5]Department of Physiology and Biophysics, School of Medicine, University of Washington, Seattle, WA, USA

Correspondence: m.kavallaris@ccia.unsw.edu.au

as generating the most dynamic microtubules, promoting microtubule catastrophe and conferring resistance to the stabilizing effects of tubulin-targeted agents both in cell-free systems (Banerjee et al, 1994, 1997; Panda et al, 1994; Derry et al, 1997; Pamula et al, 2016; Vemu et al, 2017) and in the more complex intracellular environment using forced genetics approaches (Goncalves et al, 2001; Hari et al, 2003; Kamath et al, 2005; Gan et al, 2010), although these findings have not been unequivocal (Blade et al, 1999; Gan et al, 2010; Vemu et al, 2016). Aberrant expression of this isotype in a variety of cancers is associated with resistance to tubulin-targeted agents, underscoring the importance of this particular tubulin isotype in regulating microtubule dynamics (Kavallaris, 2010; Parker et al, 2014). However, the importance of tubulin isotypes in spatially coordinating the dynamics of microtubules within the cell remains unaddressed.

Studies using cell-free systems of isolated tubulin have identified that the tubulin C-terminal tail intrinsically destabilizes microtubules and that its anionic character mediates these effects (Mejillano & Himes, 1991; Mejillano et al, 1992). Conversely, a more recent study using purified tubulin suggests that the residues within the βIIb- or βIII-tubulin body, rather than the C-terminal tail, are responsible for conferring isotype-specific effects on microtubule dynamics in vitro (Pamula et al, 2016). In silico modeling approaches have suggested that the flexible C-terminal tail peptides transit a large conformational space and interact with neighboring tubulin proteins, altering the stability and conformation of tubulin heterodimers within microtubule protofilaments (Freedman et al, 2011). However, whether the tubulin C-terminal tail plays an important role in vivo where MAPs and spatially regulated interactions contribute to microtubule dynamics remains to be resolved.

Removal of the C-terminal tail regions using subtilisin protease treatment has demonstrated the importance of the tubulin C-terminal tail region in regulating the interaction of tubulin isotype mixtures with MAPs that regulate microtubule dynamics. The mitotic centromere-associated kinesin (MCAK/Kif2C, hereafter referred to as MCAK) interacts with microtubules in the absence of the α- and β-tubulin C-terminal tails, but the tubulin C-terminal tail is required for its microtubule depolymerization activity in reduced in vitro models (Moores et al, 2002; Niederstrasser et al, 2002; Helenius et al, 2006; Hertzer & Walczak, 2008). Because MCAK is a potent inducer of microtubule catastrophe with the potential to influence microtubule network remodeling, an effect of specific tubulin isotypes on MCAK activity would have profound consequences for cellular processes that rely on the microtubule network (Montenegro Gouveia et al, 2010; Gardner et al, 2011). However, how individual tubulin isotypes regulate its activity has not yet been explored.

Although yielding important insights, cell-free approaches fail to preserve the endogenous regulatory environment afforded by the interaction of tubulin with the endogenous network of regulatory factors and forced genetics approaches perturb the tubulin balance that is a tightly regulated component of the microtubule cytoskeleton and these factors contribute to discrepancies regarding the isotype-specific effects of the tubulin isotypes. These confounding effects highlight the need for biologically relevant models with which to define the contribution of tubulin isotypes and their C-terminal tail regions in the regulation of microtubule

dynamics. Through the development of novel syngeneic human cell models that preserve the endogenous microtubule network and eliminate the endogenous unmodified βIII-tubulin protein, we reveal that the βI- and βIII-tubulin C-terminal tails spatially regulate the coordination of microtubule dynamics in an isotype-specific manner. The βIII-tubulin C-terminal tail promotes microtubule assembly and inhibits explorative microtubule extension. It decreases the sensitivity of microtubules to the microtubule-stabilizing agent paclitaxel compared with the βI-tubulin C-terminal tail and increases the sensitivity of microtubules to MCAK-mediated microtubule depolymerization. Overall, we find that the β-tubulin C-terminal tail fine-tunes and coordinates microtubule dynamics across the intracellular microtubule network in an isotype-specific manner to support critical cell functions.

## Results

### Development of gene-edited cell models expressing modified β-tubulin proteins

The β-tubulin isotype composition is recognized as an important regulator of microtubule dynamics. As the most variable region of the β-tubulin sequence, the β-tubulin C-terminal tail is hypothesized to contribute to the regulation of microtubule dynamics. However, the importance of this C-terminal tail region in regulating microtubule dynamics has not been established. We sought to address this by developing novel human cell models where the minor tubulin isotype, βIII-tubulin, was replaced with C-terminal modified versions of the protein, thereby removing the confounding influence of endogenous unmodified protein while preserving the endogenous microtubule network. NCI-H460 cell lines, which endogenously express both βI- and βIII-tubulin isotypes, were gene-edited using zinc-finger nucleases targeted to the *TUBB3* locus, which encodes the endogenous βIII-tubulin protein, to replace the endogenous βIII-tubulin protein with expression of either the full-length βIII-tubulin protein (ZB3), βIII-tubulin lacking the C-terminal tail (truncated at Thr429) (ZB3Δ), or a modified form of the protein where the βIII-tubulin C-terminal tail sequence from Thr429 was substituted for the βI-tubulin C-terminal tail sequence (ZB3/CB1) (Fig 1A). The βIII-tubulin isotype has a highly restricted distribution, where it is a minor component of the total tubulin pool, whereas the βI-tubulin C-terminal tail is ubiquitously expressed, such that modification of these tail sequences may reveal functional consequences that are relevant to tissue- and disease-specific microtubule dynamics. Because of the large *TUBB3* gene size, its poorly defined regulatory elements (Dennis et al, 2002), and the homology of β-tubulin family exonic sequences, the gene editing strategy was designed to target the 5' end of intron 1 of the *TUBB3* gene and drive expression of the modified proteins from a CMV promoter to ensure specific targeting of this isotype alone. The modified proteins were GFP tagged to enable identification of the small proportion of cells in which homologous recombination was successful and enable tracking of the modified proteins in vitro. The C-terminal tag design was conservative, consisting of a GFP tag that has been extensively validated in numerous studies to not affect microtubule dynamics (Straight et al, 1997; Ludin &

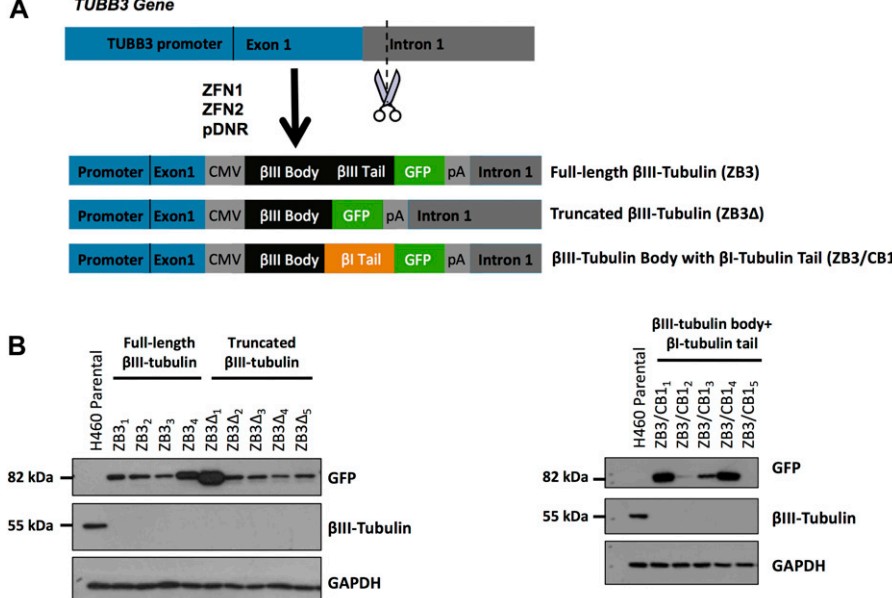

**Figure 1. Gene-edited NCI-H460 cells expressing βIII-tubulin modified at the C-terminal tail region.**
**(A)** Schematic outlining the gene editing approach to replace the endogenous βIII-tubulin protein with either the full-length βIII-tubulin protein with GFP tag (ZB3), the βIII-tubulin protein truncated at the C-terminal tail with GFP tag (ZB3Δ), or the βIII-tubulin body with a βI-tubulin C-terminal tail sequence with GFP tag (ZB3/CB1); ZFN1 and ZFN2: plasmids encoding zinc-finger nucleases; pDNR: donor cassette plasmid encoding the insertion sequence. **(B)** Representative Western blot of the gene-edited clones, which have knocked out expression of the endogenous βIII-tubulin protein (55 kD) and expression of the higher molecular weight GFP-tagged modified βIII-tubulin proteins (~82 kD). NCI-H460 parental cells, which endogenously express βIII-tubulin, are presented as a control. These Western blots are replicated in Fig S1C to measure the isotype composition of the gene-edited clones.

Matus, 1998; Heidemann et al, 1999; Rusan et al, 2001; Gan et al, 2010) and separated from the β-tubulin C-terminal tail by a 14–amino acid flexible linker to minimize steric interactions between the GFP and microtubule lattice.

From more than 500 single-cell clones screened for each structural modification to the β-tubulin C-terminal tail, between four and five clones were identified to have knocked out expression of the endogenous βIII-tubulin protein and expression of the higher molecular weight–modified βIII-tubulin protein (Fig 1B). The panel of clones generated had a range of expression levels of the modified βIII-tubulin proteins (Fig 1B). Two sets of clones with matched protein and mRNA expression levels (Set 1: $ZB3_1$, $ZB3_2$, and $ZB3/CB1_2$; Set 2: $ZB3_4$, $ZB3Δ_1$, and $ZB3/CB1_3$) representing 30% and 100% of the protein expression level of the endogenous βIII-tubulin protein in the parental cell line, respectively (Fig S1A and B), were chosen for subsequent assays examining the importance of the β-tubulin C-terminal tail in regulating microtubule dynamics while minimizing the influence of βIII-tubulin expression levels on these processes (Gan et al, 2010).

Analysis of the β-tubulin isotype composition by Western blotting indicated that modification of the C-terminal tail region did not alter the total β-tubulin levels or the β-tubulin isotype composition in these cells (Fig S1C). This concurs with findings that suppression of βIII-tubulin expression does not alter the β-tubulin isotype composition in human non–small cell lung cancer cells (Gan et al, 2007; Mccarroll et al, 2010).

### Modification of βIII-tubulin structural domains does not affect microtubule incorporation or architecture

Live- and fixed-cell microscopy was performed to determine if the modifications to the β-tubulin structural domains affected the microtubule architecture. Live-cell microscopy clearly showed that the GFP-labeled fusion proteins incorporated into the microtubule cytoskeleton in both interphase and mitotic cells (Fig 2A). Loss (ZB3Δ)

or substitution (ZB3/CB1) of the βIII-tubulin C-terminal tail sequence did not affect the incorporation of the modified tubulin proteins into the microtubule network in either interphase microtubules or spindle microtubules (Fig 2A). This was further confirmed by quantifying the level of the total and modified tubulin proteins present in polymerized form in the cell, which also demonstrated that the modified tubulin proteins in the gene-edited clones had a similar polymerization level compared with the endogenous βIII-tubulin protein in the parental cell line (Fig S2A–C). Furthermore, these modifications did not qualitatively alter the architecture of the microtubule network compared with the NCI-H460 parental cell line (Fig 2A). The absence of any abnormalities in spindle morphology is supported by the observation that loss (ZB3Δ) or substitution (ZB3/CB1) of the βIII-tubulin C-terminal tail sequence did not affect the proliferation rates of these cells as measured by BrdU incorporation (Fig 2B) and growth curve analysis using trypan blue dye exclusion and cell counting over 96 h (Fig S2D).

The observed incorporation of these modified proteins into the microtubule network strongly indicated that the structural modifications to the β-tubulin C-terminal tail region or the addition of the C-terminal GFP tag did not alter the fold of the tubulin proteins. Microtubules are highly intolerant of deviations from the native tubulin fold and adoption of the correct tubulin fold is a prerequisite of tubulin heterodimer formation and its subsequent polymerization into microtubules (Tian et al, 1999). These findings indicate that the β-tubulin C-terminal tail is not necessary for the incorporation of the β-tubulin proteins within the microtubule network in concordance with previous studies (Joe et al, 2009; Pamula et al, 2016).

### The β-tubulin C-terminal tail spatially modulates microtubule assembly rates in an isotype-specific manner

Tubulin partitioning between the soluble and polymerized fractions operates at nonequilibrium conditions to exert precise control over

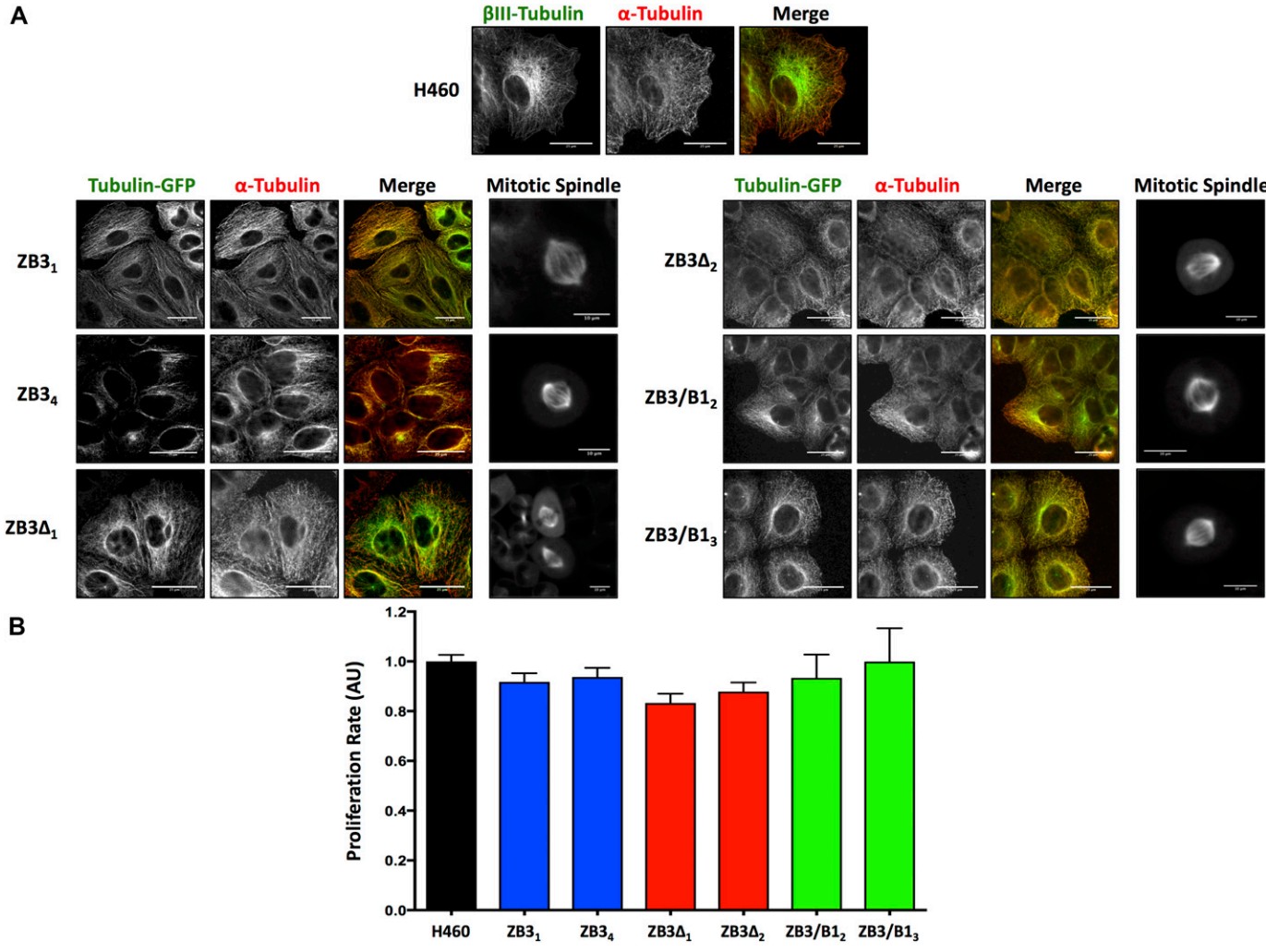

**Figure 2. Modification of the βIII-tubulin C-terminal tail does not affect the incorporation of the protein into the microtubule network, the microtubule architecture, or the cellular proliferation rate.**

**(A)** Representative immunofluorescence and live-cell images of the microtubule architecture in gene-edited NCI-H460 cell clones expressing the full-length βIII-tubulin protein (ZB3), truncated βIII-tubulin protein (ZB3Δ), or βIII-tubulin body with the βI-tubulin tail (ZB3/CB1) compared with the NCI-H460 parental cell line (H460). For interphase microtubule architecture, immunofluorescence staining was performed for α-tubulin detection. Individual channels are presented as grayscale images and the merged image with each channel colored accordingly. Left panel: modified βIII-tubulin proteins; center panel: α-tubulin; right panel: merged image of modified βIII-tubulin proteins (green) and α-tubulin (red). Scale bar 25 μm; and far right panel: higher magnification images of the mitotic spindle in live cells (GFP imaging). Scale bar 10 μm. Representative of three independent experiments. **(B)** Proliferation rates of gene-edited clones as measured by the BrdU assay and normalized to cell number. Mean ± SEM of four independent experiments, no significant difference.

microtubule assembly and disassembly events (Tian et al, 1997). Tubulin polymerization assays indicated that loss or substitution of the βIII-tubulin C-terminal tail did not affect the proportion of tubulin proteins partitioning between the soluble and polymerized fractions (Fig S3A).

To examine the importance of the β-tubulin C-terminal tail in the dynamic assembly properties of microtubules, gene-edited cells expressing modified β-tubulin proteins were transiently transfected with mCherry-labeled EB3 proteins, which track to the growing plus-end tips of assembling microtubules and can be tracked by live-cell spinning disk fluorescent microscopy (Akhmanova & Steinmetz, 2008; Maurer et al, 2012). The speed of EB3 comets was measured as a readout for the assembly rate of microtubules, and whole-cell analysis was performed using the TrackMate particle tracking

algorithms (Fig 3A). In contrast to alternative approaches, this whole-cell analysis approach minimizes the bias introduced when tracking microtubules only at the cell periphery, where the microtubule assembly rates are reduced (Waterman-Storer et al, 2000; Plestant et al, 2014).

The microtubule assembly parameters of cell lines expressing the full-length βIII-tubulin GFP-tagged protein (ZB3) did not differ significantly from that of the parental cell line (Fig S3B), confirming that the GFP tag does not interfere with microtubule dynamics as observed by others (Straight et al, 1997; Ludin & Matus, 1998; Heidemann et al, 1999; Rusan et al, 2001; Gan et al, 2010) and validating the use of these gene-edited cell models to study the role of the β-tubulin C-terminal tail in regulating microtubule dynamics.

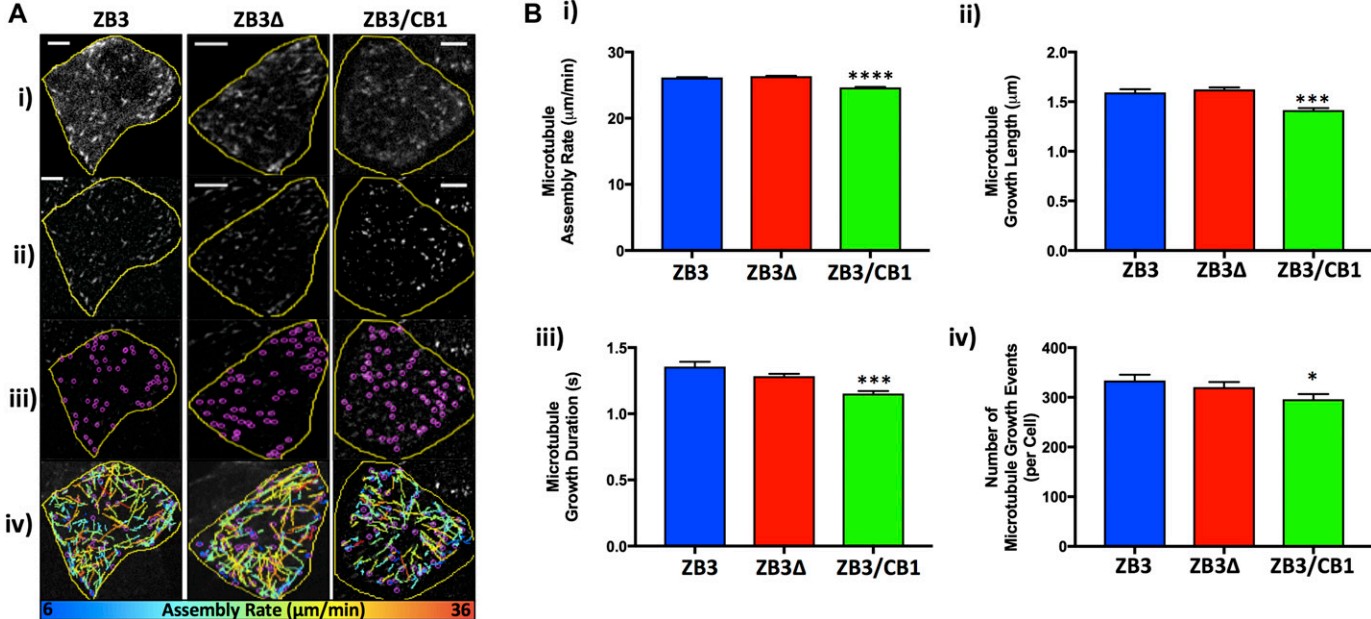

**Figure 3. The β-tubulin C-terminal tail modulates microtubule assembly in an isotype-dependent manner.**
**(A)** Representative images of EB3-transfected gene-edited cells (i). The image processing involved in measuring the assembly parameters: image appearance following the processing of raw images (ii), identification of microtubule plus ends by the particle tracking analysis (iii), and microtubule assembly events colored according to assembly rate (iv). Scale bar 5 μm. **(B)** Microtubule assembly parameters in gene-edited NCI-H460 cells expressing the full-length βIII-tubulin protein (ZB3), truncated βIII-tubulin protein (ZB3Δ), and βIII-tubulin body with βI-tubulin tail (ZB3/CB1) as measured by EB3-mCherry motion and particle tracking. The microtubule assembly rate (i), microtubule growth length (ii), microtubule growth duration (iii), and the number of microtubule growth events (iv) were calculated as the average value per cell and are presented as the per-cell mean ± SEM of at least 50 cells in each of three independent experiments for each tubulin modification. *$P < 0.05$, ***$P < 0.001$, and ****$P < 0.0001$ relative to cells expressing the full-length protein (ZB3).

In normal growth conditions, loss of the βIII-tubulin C-terminal tail (ZB3Δ) did not significantly affect the microtubule assembly rate (Fig 3A and Bi), growth length (Fig 3Bii), duration of growth events (Fig 3Biii), or the number of growing microtubules per cell (Fig 3Biv), compared with cells expressing the full-length βIII-tubulin protein (ZB3; Fig 3A and B, and Table S1). However, substitution of the βIII-tubulin tail sequence with the βI-tubulin tail sequence (ZB3/CB1) did significantly reduce these measures of microtubule assembly compared with cells expressing the full-length βIII-tubulin protein (ZB3; Fig 3B and Table S1). This indicates that the β-tubulin C-terminal tail region regulates the rate and propensity of microtubule assembly in an isotype-specific manner.

For the microtubule network to support fundamental cellular functions, the dynamic behavior of individual microtubules must be coordinated across the cytoplasm. To investigate how the tubulin C-terminal tail affects the coordinated assembly of microtubules, high-resolution live-cell images were captured and analyzed by spatiotemporal image cross-correlation spectroscopy (STICCS). STICCS resolves the directed velocity of fluorescently labeled proteins across the cell from time series images without requiring particle tracking procedures (Hebert et al, 2005) (Fig S4A). Using this method, cross-correlation analysis of images from two different fluorescent channels can be used to measure the cotransport of the GFP-labeled tubulin and the red fluorescent protein (RFP)-labeled EB3 (Toplak et al, 2012) (Figs 4 and S4A), thereby revealing not only the velocities of microtubule movement and assembly but also the relationship between initiated microtubule assembly events and existing microtubule tracks. These analyses across multiple cells

confirmed the findings of the particle tracking analysis that substitution of the βIII-tubulin C-terminal tail with the βI-tubulin C-terminal tail (ZB3/CB1) significantly reduced the microtubule assembly rate (Figs 4A and S4Bii) and number of assembly events (Figs 4A and S4Biii) compared with cells expressing the full-length protein (ZB3). The overall lower microtubule assembly rate measured by this method compared with particle tracking analysis is due to the ability of this method to measure slow microtubule growth events that are not robustly trackable by particle tracking methods. The loss (ZB3Δ) or substitution (ZB3/CB1) of the βIII-tubulin C-terminal tail region significantly decreased the speed of tubulin movement (ZB3: 22.96 ± 2.36 μm/min; ZB3Δ: 13.7 ± 1.06 μm/min; and ZB3/CB1: 5.82 ± 1.11 μm/min; Figs 4A and S4Bi), further supporting the assertion that the βIII-tubulin C-terminal tail promotes microtubule dynamicity.

This high-resolution imaging visualized microtubule growth events where the microtubule plus end grows toward unchartered cytoplasmic space, that is, explorative microtubule growth (Fig 4Bi) and the assembly of microtubules along existing microtubule fibers (Fig 4Bii). STICCS cross-correlation distinguishes between these scenarios, with the coregistered growth of microtubule fibers with the EB3-labeled tip registered as a cross-correlated growth event, whereas the growth of microtubules along existing microtubule fibers represent non–cross-correlated microtubule growth (Fig 4B). Therefore, STICCS analysis measures the collective spatial behavior of microtubule growth events that cannot be discerned by particle tracking methods. Cross-correlation analysis indicated that the growth of microtubules along new paths (i.e., explorative microtubule growth) occurs at a higher speed than

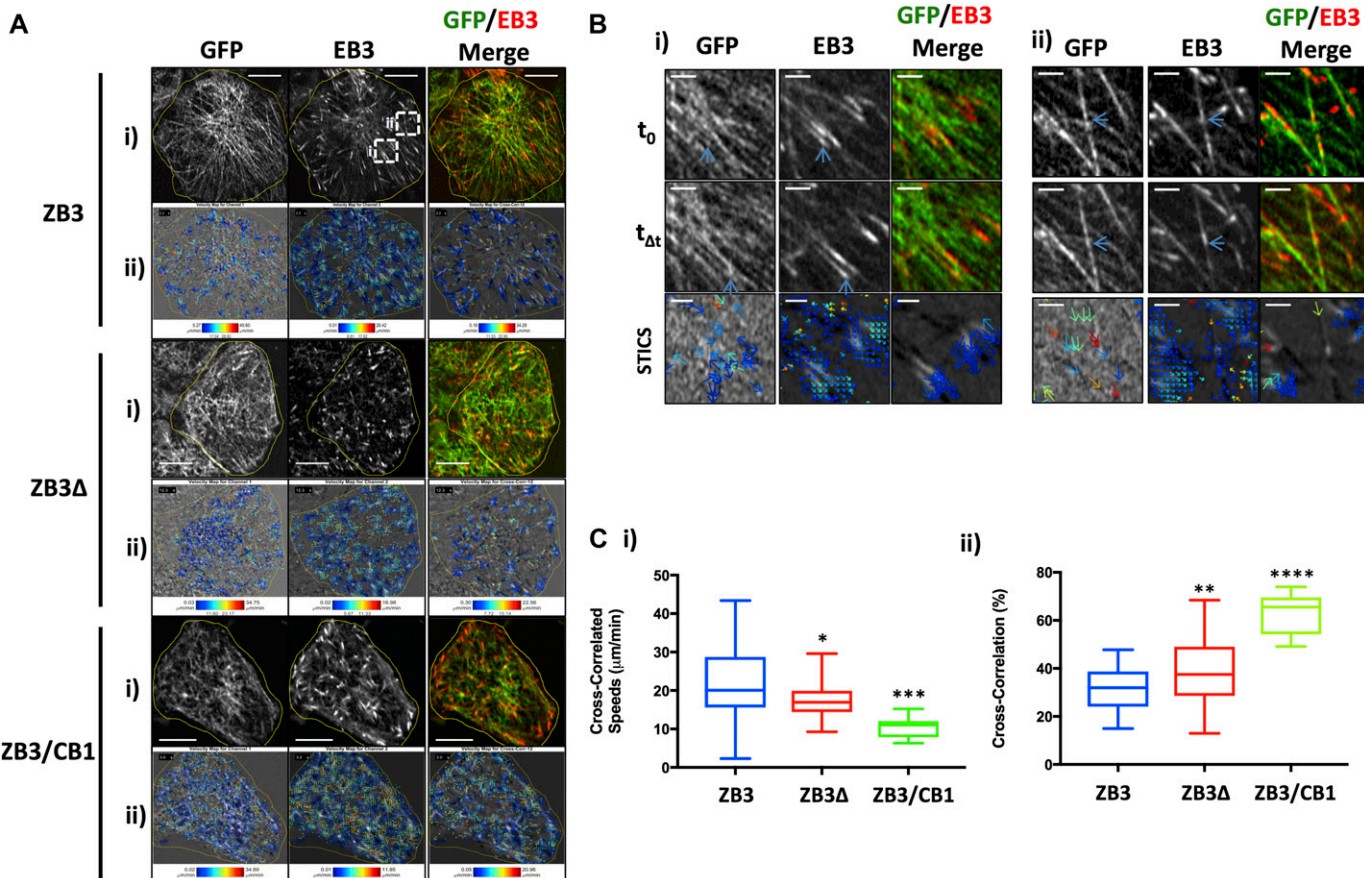

**Figure 4.  The β-tubulin C-terminal tail modulates the coordination of microtubule assembly in an isotype-dependent manner.**
**(A)** Representative images and STICCS analysis of gene-edited NCI-H460 cells expressing the full-length βIII-tubulin protein (ZB3), truncated βIII-tubulin protein (ZB3Δ), and βIII-tubulin body with βI-tubulin tail (ZB3/CB1). (i) Raw images of tubulin-GFP (left) and EB3-RFP (center) and merged image (right) at a single time point. (ii) STICCS vector maps showing the velocity of the movement in a single time block. The direction of the arrows indicates the direction of movement of the tubulin-GFP (left), EB3-RFP (center), and cross-correlation of GFP and EB3 (right); and the color indicates the speed of movement. Scale bar 5 μm. **(B)** Representative close-up images of the regions of the ZB3 cells indicated by the white boxes for microtubule and EB3 movement between the time t0 and Δt for the GFP (left panel), EB3 (center panel), and merged (right panel) channel showing microtubule assembly that do (i) and do not (ii) correlate with tubulin movement, together with the STICCS vector maps corresponding to these ROI and TOI regions (lower panel) for the GFP autocorrelation (left panel), EB3 autocorrelation (center panel), and cross-correlation between GFP and EB3 (right panel). Blue arrows indicate microtubule assembly events that are (i) and are not (ii) correlated. Microtubule assembly along existing microtubule fibers (ii) is shown as noncorrelating microtubule assembly. Conversely, explorative microtubule growth where the GFP and EB3 signals proceed at the similar rate into new territory (ii) is measured as a correlated STICCS event (i). Scale bar 2 μm. **(C)** Microtubule assembly dynamics as measured by STICCS, showing the speed of cross-correlated movement between the microtubule and EB3 channels (i) and the proportion of microtubule assembly events that are cross-correlated with the movement of microtubules (ii). Graphs give the median, box gives the 25th to the 75th percentile, and whiskers give the minimum and maximum values of at least 10 cells from two independent experiments. *P < 0.05, **P < 0.01, ***P < 0.001, and ****P < 0.0001 relative to cells expressing the full-length βIII-tubulin protein. Corresponding values are presented in Table S1.

the average microtubule assembly rate (Fig 4Ci). Conversely, microtubule assembly along existing microtubule fibers occurs at a slower speed. Importantly, loss of the βIII-tubulin C-terminal tail region significantly reduced the speed of these explorative assembly events (ZB3Δ: 17.26 ± 0.78 μm/min) compared with microtubules decorated with the βIII-tubulin C-terminal tail (ZB3: 22.33 ± 2.02 μm/min) (Fig 4Ci). This decrease was further pronounced when the βIII-tubulin C-terminal tail was substituted for the βI-tubulin C-terminal tail (ZB3/CB1: 10.51 ± 0.83 μm/min) (Fig 4Ci). Loss or substitution of the βIII-tubulin C-terminal tail also significantly increased the proportion of assembly events occurring as explorative growth events (ZB3: 31.37 ± 1.64%; ZB3Δ: 39.46 ± 1.85%; and ZB3/CB1: 62.98 ± 2.77% (Fig 4Cii). Collectively, this indicates that compared with the βIII-tubulin C-terminal tail, the βI-tubulin C-terminal tail promotes explorative microtubule assembly along new paths but

that these occur at a slower speed, thereby decreasing the average microtubule assembly rate (Fig 3). Therefore, the β-tubulin C-terminal tail regulates the collective behavior of microtubules that are in close proximity and in doing so alters the spatial distribution of the microtubule assembly rate across the microtubule network.

### The β-tubulin C-terminal tail sequence modulates microtubule assembly in response to paclitaxel

Previous studies have highlighted that the β-tubulin isotype composition, and particularly the βIII-tubulin expression levels, differentially influence the effects of tubulin-binding agents on microtubule dynamics (Kamath et al, 2005; Gan et al, 2010). Therefore, the importance of the tubulin C-terminal tail regions on

tubulin partitioning and microtubule assembly parameters in response to microtubule-stabilizing agents was examined by treatment of gene-edited cells with paclitaxel. Low and intermediate concentrations (6–20 nM) of paclitaxel were used to recapitulate physiologically relevant doses at which the presence of the βIII-tubulin protein modulates microtubule dynamics, but cell viability and cell cycle progression are inhibited only after prolonged exposure to this tubulin-targeted agent (Gan et al, 2010).

Whereas paclitaxel treatment increased the amount of polymerized tubulin, as expected, modification of the β-tubulin C-terminal tail region did not significantly affect the partitioning of tubulin between the soluble and polymerized fractions (Fig S5). In cells expressing the full-length βIII-tubulin protein (ZB3), paclitaxel treatment significantly reduced the microtubule assembly rate (Fig 5Ai), microtubule growth length (Fig 5Aii), and number of assembly events (Fig 5Aiv and Table S1), consistent with previous studies (Kamath et al, 2005; Gan et al, 2010). Treatment with paclitaxel increased the microtubule growth duration (Fig 5Aiii), consistent with the effects of this drug in suppressing catastrophe rates at these concentrations (Gan et al, 2010). Of these measures of microtubule assembly, paclitaxel treatment induced a more profound decrease in the number and length of microtubule assembly events, with a lesser effect on the microtubule assembly rate.

In the presence of paclitaxel, loss of the βIII-tubulin C-terminal tail (ZB3Δ) significantly decreased microtubule growth length (Fig 5Aii) and number of assembling microtubules (Fig 5Aiv and Table S1). However, in these conditions, loss of the βIII-tubulin C-terminal tail rendered microtubules more resistant to the effects of paclitaxel in decreasing the microtubule assembly rate (Fig 5Ai) and increasing the microtubule growth duration (Fig 5Aiii and Table S1). These effects were more pronounced when the βIII-tubulin C-terminal tail was replaced with the βI-tubulin C-terminal tail (ZB3/CB1; Fig 5A and Table S1), suggesting that compared with the βIII-tubulin C-terminal tail, the βI-tubulin C-terminal tail increases the sensitivity of microtubules to spatial effects of paclitaxel on the number and length of assembly events but ameliorates the effects of paclitaxel on the assembly rate and growth duration.

STICCS further revealed that paclitaxel treatment significantly reduced the proportion of cross-correlated assembly events, that is, explorative assembly events in which microtubule growth is driven into unchartered cytoplasmic space, compared with assembly alongside an existing microtubule fiber (Figs 5Bi and 4B). This was accompanied by an increase in the speed of these explorative assembly events compared with microtubules in untreated cells (Fig 5Bii). In particular, by 2 h of treatment, paclitaxel induced approximately a two-fold increase in the speed of these explorative assembly events for all tubulin C-terminal tail modifications, such that their assembly rate remained significantly lower in cells where the βIII-tubulin C-terminal tail was substituted for the βI-tubulin C-terminal tail (ZB3/CB1: 22.07 ± 1.74 μm/min) compared with cells expressing the full-length (ZB3: 43.39 ± 4.43 μm/min) or truncated βIII-tubulin proteins (ZB3Δ: 34.14 ± 5.29 μm/min) (Fig 5Bii). Importantly, compared with microtubules decorated with the full-length βIII-tubulin protein (ZB3: 24.31 ± 1.7% decrease), those decorated with the βI-tubulin C-terminal tail in place of the βIII-tubulin C-terminal tail suffered a significantly greater reduction in the proportion of explorative microtubule

assembly events (ZB3/CB1: 41.1 ± 3.2% decrease) on treatment with paclitaxel (Fig 5Bi), suggesting that the βI-tubulin C-terminal tail renders microtubules more sensitive to paclitaxel's suppression of explorative microtubule assembly. Together, this suggests that the tubulin C-terminal tail region differentially confers sensitivity to the effect of paclitaxel in suppressing the microtubule assembly rate and the propensity of microtubules to initiate and maintain a microtubule growth state as well as the coordinated growth of microtubules along new paths. Although the βI-tubulin C-terminal tail maintains higher average microtubule assembly rates than the βIII-tubulin C-terminal tail in the presence of paclitaxel, it increases the susceptibility of the microtubule to stabilization and to paclitaxel-induced microtubule coupling, thereby profoundly perturbing the spatial distribution of microtubule dynamics in response to this agent.

## The βIII-tubulin C-terminal tail confers sensitivity to MCAK-mediated microtubule depolymerization in an isotype-specific manner

Like microtubule assembly, microtubule disassembly is also regulated by the tubulin isotype composition and an extensive network of MAPs. The MCAK is a nonmotile kinesin that induces tubulin depolymerization in a manner that is dependent on the tubulin C-terminal tail (Moores et al, 2002; Niederstrasser et al, 2002; Helenius et al, 2006; Hertzer & Walczak, 2008). However, how the isotype composition affects the activity of this MAP is unknown.

To investigate the importance of the β-tubulin C-terminal tail on the microtubule depolymerizing activity of MCAK, two sets of expression-matched gene-edited cells expressing modified β-tubulin proteins were transiently transfected with MCAK-mCherry, and the activity of MCAK was assessed by immunofluorescence staining of the microtubule network. Depolymerization of the microtubule network by MCAK reduces the level of polymerized tubulin and, therefore, the α-tubulin fluorescence intensity after soluble tubulin heterodimers is removed by the fixation and washing protocol.

Although comprising only a minor fraction of the total tubulin pool (Nicoletti et al, 2001), βIII-tubulin was found to significantly contribute to the resistance of microtubules to MCAK-mediated depolymerization because suppression of βIII-tubulin expression by stable expression of a βIII-tubulin-targeted shRNA (Mccarroll et al, 2010) significantly increased the microtubule depolymerizing activity of MCAK compared with control cells (Ctrl$_{SH2}$: 0.690 ± 0.033 AU and βIII$_{SH4}$: 0.393 ± 0.015 AU) (Fig S6A). However, loss of only the C-terminal tail region of the βIII-tubulin protein did not consistently recapitulate this effect in both sets of expression-matched clones (Fig 6, open versus striped bars). Rather, substitution of the βIII-tubulin C-terminal tail region with the βI-tubulin C-terminal tail profoundly increased the resistance of microtubules to the microtubule depolymerizing activity of MCAK (ZB3/CB1$_2$: 0.989 ± 0.022 AU and ZB3/CB1$_3$: 1.332 ± 0.046 AU) compared with microtubules decorated with the βIII-tubulin C-terminal tail sequence (ZB3$_1$: 0.691 ± 0.018 AU and ZB3$_4$: 0.649 ± 0.016 AU) (Fig 6). Therefore, the β-tubulin C-terminal tail regions impact the microtubule depolymerizing activity of MCAK in an isotype-dependent manner.

Similarly, in CHO cells, which lack endogenous expression of the βIII-tubulin protein (Sawada & Cabral, 1989), the βIII-tubulin C-terminal tail significantly increased the sensitivity of microtubules to

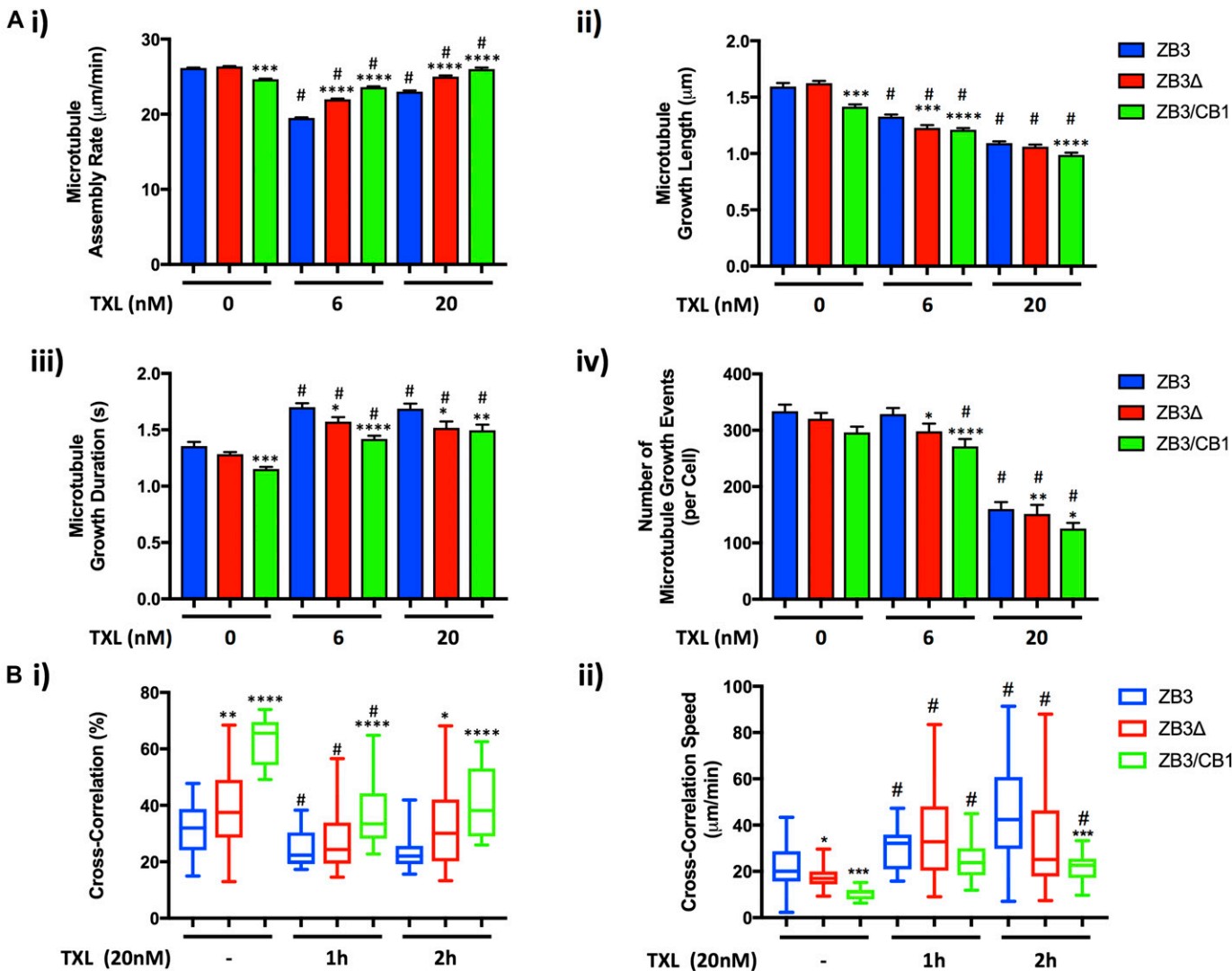

**Figure 5. The β-tubulin C-terminal tail modulates the sensitivity of microtubules to paclitaxel.**
**(A)** Microtubule assembly parameters in gene-edited NCI-H460 cells expressing either the full-length βIII-tubulin protein (ZB3), truncated βIII-tubulin protein (ZB3Δ), or βIII-tubulin body with βI-tubulin C-terminal tail (ZB3/CB1) cultured in normal growth media as measured by EB3-mCherry motion and particle tracking in response to paclitaxel treatment (TXL, 6 or 20 nM for 2 h). The microtubule assembly rate (i), microtubule growth length (ii), microtubule growth duration (iii), and number of microtubule growth events (iv) are presented as the mean ± SEM of at least 50 cells in each of three independent experiments for each tubulin modification. $*P < 0.05$, $***P < 0.001$, and $****P < 0.0001$ relative to cells expressing the full-length protein (ZB3); $^{\#}P < 0.05$ relative to untreated cells expressing the same type of tubulin modification. **(B)** Microtubule assembly dynamics as measured by spatiotemporal image correlation spectroscopy for cells treated with paclitaxel (20 nM for 1 or 2 h), showing the speed of cross-correlated movement between the microtubule and EB3 channels (i) and the proportion of microtubule assembly events that are cross-correlated with the microtubule movement (ii). Graphs give the median, box gives the 25th to the 75th percentile, and whiskers give the minimum and maximum values of at least 15 cells from two independent experiments. $*P < 0.05$, $**P < 0.01$, $***P < 0.001$, and $****P < 0.0001$ relative to cells expressing the full-length βIII-tubulin protein; $^{\#}P < 0.05$ relative to untreated cells expressing the same type of tubulin modification. Measurements of untreated cells are reproduced from Fig 3. Corresponding values are presented in Table S1.

MCAK-mediated depolymerization, as indicated by decreased MCAK activity in cells expressing the truncated βIII-tubulin protein (B3Δ) compared with cells expressing the full-length βIII-tubulin protein (B3; Fig S6B). However, in contrast to the findings in NCI-H460 cells, this effect was not further exacerbated, but instead could be sufficiently compensated for, by the βI-tubulin C-terminal tail (B3/CB1; Fig S6B). To underscore this observation, we found that addition of the βIII-tubulin tail to the βI-tubulin body significantly increased susceptibility to MCAK (B1/CB3; Fig S6B).

The tubulin C-terminal tail modifications did not affect the level of polymerized tubulin when cells were treated with the destabilizing agent nocodazole (Fig S7A), indicating that the higher activity of MCAK on microtubules containing the βIII-tubulin C-terminal tail is not because of intrinsic differences in the stability of microtubules induced by these tubulin modifications. Similarly, levels of tyrosinated tubulin, which promote MCAK activity (Peris et al, 2009), were also unaltered by tubulin C-terminal tail modifications (Fig S7B), indicating that the differential effect of MCAK on microtubules decorated by the βIII-tubulin or the βI-tubulin C-terminal tail is directly related to MCAK activity rather than being a result of altered tubulin posttranslational modifications. Collectively, these studies demonstrate that the β-tubulin

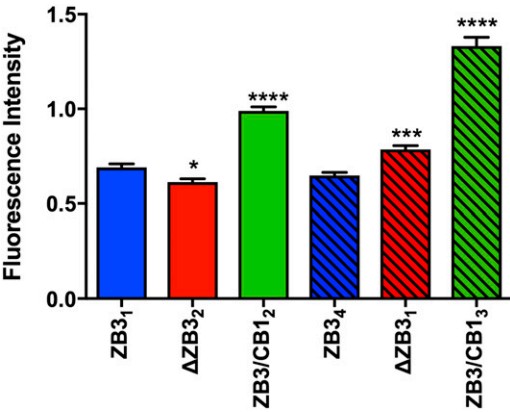

**Figure 6. The βI-tubulin C-terminal tail confers resistance to MCAK-mediated microtubule depolymerization.**
**(A)** MCAK activity in two sets of expression-matched gene-edited NCI-H460 cells as measured by normalized α-tubulin fluorescence. Mean ± SEM of at least 100 cells of each type in each of three independent experiments. Solid and striped bars indicate different sets of expression-matched gene-edited clones. *$P < 0.05$, ***$P < 0.001$, and ****$P < 0.0001$ relative to cells expressing the full-length βIII-tubulin protein.

C-terminal tail modulates microtubule dynamics in normal growth conditions and in response to stabilizing and destabilizing factors.

# Discussion

The dynamics of the microtubule cytoskeleton is highly spatially and temporally regulated to support fundamental cellular processes, although the mechanisms that orchestrate microtubule dynamics remain poorly defined. The different tissue distributions of the tubulin isotypes infer unique, but hitherto uncharacterized, functions for members of this protein family in regulating microtubule dynamics. In particular, interest in the βIII-tubulin and its contribution to microtubule dynamics has been spurred on by in vitro studies identifying its unique dynamic properties in purified microtubules (Panda et al, 1994; Derry et al, 1997), its highly restricted tissue distribution, and observations of aberrant expression in cancers that are resistant to the tubulin-binding agent class of chemotherapeutics (Kavallaris, 2010; Parker et al, 2014). The most divergent region of the β-tubulin isotypes, the C-terminal tail, represents a potential candidate for conferring isotype-specific characteristics to microtubules. However, attempts to resolve the importance of the β-tubulin C-terminal tail in regulating microtubule dynamics in human cells have been hindered by a lack of tools to accurately manipulate the protein composition while retaining the endogenous tubulin isotype composition and intracellular regulatory network that regulate microtubule dynamics. To this end, we developed syngeneic human cell models that enable measurements of microtubule dynamics while retaining the endogenous microtubule network. By using gene editing technology to alter the sequence of the endogenous βIII-tubulin gene in human cells, this approach enabled specific and precise modification and substitution of the β-tubulin C-terminal tail region while preserving the β-tubulin body sequence and eliminating the influence of the endogenous unmodified tubulin protein on microtubule

dynamics. Using these novel cell models, the outlined studies have revealed that the β-tubulin C-terminal tail is an important regulator of microtubule dynamics both in spatial and temporal space in an isotype-specific manner. In particular, compared with the βI-tubulin C-terminal tail, the βIII-tubulin C-terminal tail increases the propensity of microtubules to grow and shorten and promotes high speed microtubule assembly along existing microtubule fibers, thereby fine-tuning microtubule dynamics in a spatiotemporal manner.

The outlined studies identified that the β-tubulin C-terminal tail has a major influence on the propensity of microtubules to initialize and maintain a microtubule assembly event in human cells. This was shown by an increased number of microtubule assembly events and longer assembly events for microtubules decorated with the βIII-tubulin C-terminal tail compared with the βI-tubulin C-terminal tail in normal growth conditions and in response to pharmacological stabilization with paclitaxel. These findings concur with observations from cell-free systems and forced genetic studies that the βIII-tubulin isotype promotes microtubule dynamicity compared with other tubulin isotypes (Panda et al, 1994; Derry et al, 1997; Goncalves et al, 2001; Kamath et al, 2005; Gan et al, 2010). However, these findings are in contrast to those of Pamula et al (2016), where isotype switching the βIII-tubulin tail for the βIIb-tubulin tail in a purified tubulin in vitro system did not significantly alter the assembly rate or catastrophe frequency (Pamula et al, 2016). Furthermore, other in vitro approaches have indicated that the addition of α1a/βIII-tubulin to an α1b/βI + βIVb tubulin mixture increases the microtubule catastrophe frequency (Vemu et al, 2017), in contrast to our observations that the βIII-tubulin C-terminal tail increases the growth duration compared with the βI-tubulin C-terminal tail. These differences may stem from the fact that in vitro systems do not recapitulate the spatial coordination of microtubules and their interaction with microtubule-associated proteins that this study has identified as major isotype-specific effects conferred by the tubulin C-terminal tail. The importance of the C-terminal tail in regulating microtubule dynamics is further supported by observations in a *Caenorhabditis elegans* model that two β-tubulin isotypes, which differ by four residues distributed throughout the tubulin body and ten residues in the C-terminal tail region, differentially alter the spindle microtubule growth rate and duration in vivo (Honda et al, 2017).

Importantly, STICCS analysis has revealed the novel finding that the β-tubulin C-terminal tail affects the spatial regulation of microtubule assembly by regulating the propensity and rate of explorative microtubule assembly events that occur independently of existing proximal microtubule fibers. In particular, the βIII-tubulin C-terminal tail promotes microtubule assembly along existing microtubule fibers and promotes higher assembly velocities for explorative microtubule assembly events that occur independently of proximal microtubules. Through these effects, the spatial distribution of tubulin isotypes within a microtubule locally modulates the distribution and speed of microtubule assembly events within the cell. Although the β-tubulin C-terminal tail significantly affects the average microtubule assembly rate in an isotype-dependent manner, its influence on this parameter is less pronounced than its effects on the microtubule growth duration, length, and number of assembly events when challenged by the tubulin stabilizing agent

paclitaxel. This suggests that the β-tubulin C-terminal tails principally modulate the distribution of assembly events within the cell. This concurs with observations that individual tubulin isotypes have similar assembly rates in vitro (Rezania et al, 2008) and may account for previous studies that did not identify an effect of βIII-tubulin expression on microtubule dynamics in normal growth conditions (Gan et al, 2010). The role for the β-tubulin tail in regulating the collective behavior of microtubules provides evidence in support of the hypothesis that the local interactions between different tubulin isotypes within and between microtubules may contribute to mixing and clustering effects that result in nonlinear dynamic behavior (Panda et al, 1994; Rezania et al, 2008).

The sequence of the C-terminal tail itself contributes to paclitaxel resistance and alters the microtubule growth propensity. This information is encoded within the β-tubulin C-terminal tail sequence itself because substitution, but not loss, of the β-tubulin C-terminal tail sequence exhibited the strongest effect on microtubule assembly parameters. The C-terminal tails are likely to operate in concert with isotype-specific residues within the tubulin body (Freedman et al, 2009; Yang et al, 2016) to collectively mediate isotype-specific effects on microtubule assembly. The paclitaxel binding site is located within the tubulin body, but structural deviations in the microtubule lattice indirectly induced by the C-terminal tail may affect the ability of paclitaxel to interact with these sites (Nogales et al, 1995; Freedman et al, 2009). The unique electrostatic character of the βIII-tubulin C-terminal tail and, in particular, its terminal lysine residue may enable differential interaction of this C-terminal tail with the intermediate domain of neighboring α-tubulin subunits compared with the βI-tubulin C-terminal tail (Sherman et al, 1983; Szasz et al, 1986; Mejillano et al, 1992; Pal et al, 2001; Freedman et al, 2011; Laurin et al, 2017). In addition, molecular dynamics simulations of βI- and βIII-tubulin C-terminal tail conformations suggest that these two tails differ in their contact with the β- and α-tubulin surface residues located principally in the intermediate domains of these proteins (Downing & Nogales, 1999; Laurin et al, 2017). Whereas the C-terminal tails are too disordered to resolve by cryo-EM, microtubules composed of βIII-tubulin have subtle progressive deviations in the microtubule fiber structure (Vemu et al, 2016), and these may be influenced by interactions between the tail and tubulin body. These interactions may contribute to conformational strain in the microtubule lattice to alter the propensity of the microtubule to grow. In addition, the lack of an effect of the β-tubulin C-terminal tail sequence on soluble/polymerized tubulin partitioning does suggest that accessory interacting proteins may also contribute to conferring the isotype-specific effects of the C-terminal tail sequence on microtubule growth (Mohan et al, 2013).

The importance of the β-tubulin C-terminal tails in regulating microtubule dynamics also extends to microtubule depolymerization events. Previous studies using subtilisin-cleaved tubulin in cell-free systems showed that the tubulin C-terminal tails are required for the microtubule depolymerizing activity of the nonmotor kinesin MCAK but do not affect its ability to associate with the microtubule lattice (Moores et al, 2002; Niederstrasser et al, 2002; Helenius et al, 2006; Hertzer & Walczak, 2008), although the isotype specificity of these effects remained unexplored. The findings described here are the first to demonstrate that the β-tubulin C-terminal tail sequence regulates the microtubule depolymerizing activity of MCAK in human cells in an isotype-specific manner. In particular, the βIII-tubulin C-terminal tail increases the sensitivity of microtubules to MCAK activity compared with the βI-tubulin C-terminal tail. MCAK initiates catastrophe events and removes tubulin dimers in an ATP-dependent manner (Montenegro Gouveia et al, 2010; Gardner et al, 2011). Its ability to depolymerize microtubules depends on the affinity of the protein for the microtubule lattice, a factor that is largely governed by electrostatic interactions between MCAK's neck region and the microtubule surface charge, as well as the ability of MCAK to dissociate from released tubulin heterodimers to complete the cycle (Cooper et al, 2010). Both these mechanisms may explain the increased activity of MCAK on microtubules decorated with the βIII-tubulin C-terminal tail. Electrostatic interactions between MCAK and the tubulin heterodimers become particularly important as MCAK switches to a tightly bound depolymerization-competent state at the microtubule plus end (Friel & Howard, 2011), but must also be weak enough to enable efficient release of the tubulin heterodimer once it has been removed from the microtubule polymer. It is likely that the charge distribution, rather than simply the overall charge, across the tubulin C-terminal tails plays an important role in MCAK activity, given that the presence of a terminal tyrosine on the α-tubulin C-terminal tail enhanced MCAK processivity in cell-free systems and subtle changes in amino acid sequence within the tubulin C-terminal tail region have been shown to broadly influence kinesin processivity (Sirajuddin et al, 2014; Feizabadi, 2016). In the absence of any posttranslational modifications, the βIII-tubulin C-terminal tail carries more negatively charged residues than the βI-tubulin C-terminal tail (Roll-Mecak, 2015); however, because of its length and terminal lysine residue, the βIII-tubulin C-terminal tail has a higher pI than the βI-tubulin C-terminal tail. The higher pI of the βIII-tubulin C-terminal tail compared with the βI-tubulin C-terminal tail may enable adequate association of MCAK with the microtubule but improve the efficiency of tubulin heterodimer release each depolymerization cycle. An interaction between the positively charged C-terminal lysine of the βIII-tubulin C-terminal tail and the cationic neck region of MCAK may further facilitate more efficient release of the tubulin heterodimer from MCAK, thereby rendering microtubules more sensitive to depolymerization by this catastrophe-promoting factor. Alternatively, it is also plausible that the tubulin C-terminal tail may modulate the recruitment of microtubule-associated proteins to influence MCAK's association with, and activity at, the microtubule tip (Montenegro Gouveia et al, 2010).

Elevated microtubule assembly rates increase chromosomal instability (Ertych et al, 2014). It is possible that the faster microtubule assembly rates conferred by the βIII-tubulin C-terminal tail compared with the βI-tubulin C-terminal tail may thereby promote cellular transformation events underlying clinical correlations between high βIII-tubulin expression and aggressive and treatment refractory cancers (Kavallaris, 2010; Parker et al, 2014). Furthermore, the finding that the βIII-tubulin C-terminal tail promotes resistance to the effects of paclitaxel on the number and maintenance of assembly events provides a rational framework for the development of novel therapeutics targeted at the βIII-tubulin tail region, which may find utility in the effective treatment of paclitaxel-resistant cancers.

The gene-edited cellular models developed in this study revealed previously undefined isotype-specific roles for the βI- and βIII-tubulin C-terminal tails in spatiotemporally coordinating

microtubule dynamics within the context of a preserved microtubule network, without confounding interference from the endogenous unmodified βIII-tubulin protein and with precise structural modifications to the C-terminal tail region. However, these models are limited by the need to fluorescently tag the tubulin proteins to identify gene-edited cells and enable live-cell imaging of the microtubule dynamics. Although this tag did not affect microtubule dynamics compared with the parental cell line, it cannot be eliminated that the tag may induce some steric constraints that have a subtle influence on some protein–protein interactions with the microtubule lattice to affect other aspects of microtubule biology.

The observations that the β-tubulin C-terminal tails regulate the initiation and maintenance of microtubule growth and the sensitivity of microtubules to MCAK collectively suggest that the tubulin C-terminal tail regions modulate the energetics required for the transition of a microtubule from a paused, stable state to one of growth or disassembly (Fig 7). In particular, the βIII-tubulin

C-terminal tail primes microtubules toward growth or disassembly compared with the βI-tubulin C-terminal tail and supports a model in which the β-tubulin C-terminal tail region regulates the energy barrier associated with these transitions in dynamic state (Fig 6). The height of this barrier is lower for microtubules decorated with the βIII-tubulin C-terminal tail than the βI-tubulin C-terminal tail, thereby promoting the initiation of microtubule assembly events and sensitizing microtubules to the depolymerizing activity of catastrophe factors such as MCAK. Furthermore, by affecting the propensity for and rate of microtubule growth along microtubule fibers, or independently of surrounding microtubules, the β-tubulin C-terminal tails spatially regulate the distribution of microtubule dynamics within the cell. Whether this dynamic microtubule behavior along existing microtubule fibers contributes to lowering the energy barrier for microtubule state switching remains to be determined. Overall, the β-tubulin C-terminal tail region is an important regulator of microtubule dynamics and contributes to

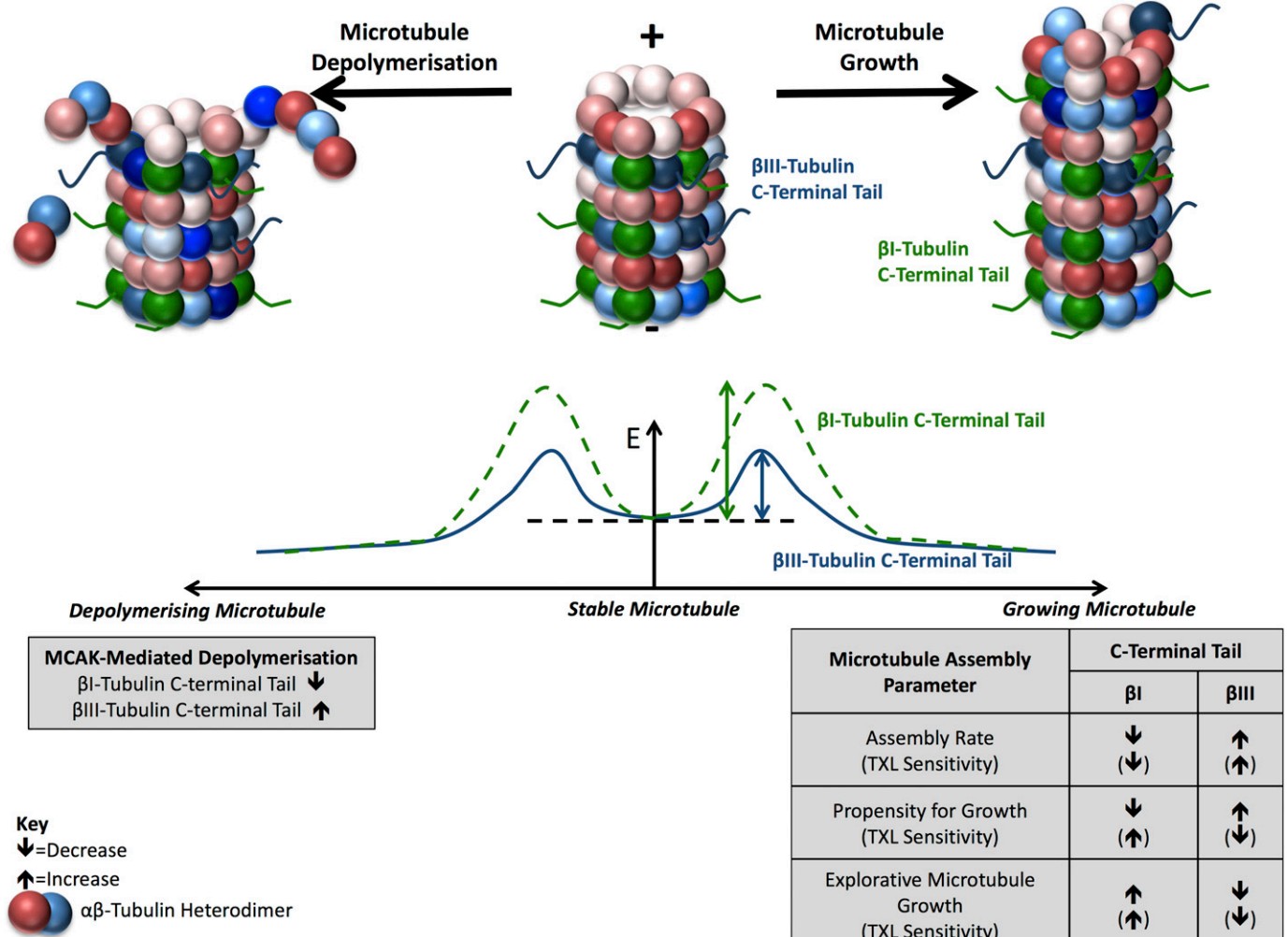

**Figure 7. The β-tubulin C-terminal tail region spatiotemporally regulates microtubule dynamics in an isotype-dependent manner.**
Schematic of a model summarizing findings that the β-tubulin C-terminal tails regulate microtubule growth and assembly. Compared with the ubiquitous βI-tubulin C-terminal tail, the βIII-tubulin C-terminal tail increases the propensity of microtubules to grow and depolymerize in normal growth conditions and under the influence of MCAK or microtubule-stabilizing agents. The C-terminal tail region also modulates the collective dynamics of microtubules such that the β-tubulin isotype C-terminal tails confer unique dynamic properties to microtubules. E, Energy; TXL, paclitaxel; ↑ and ↓, increased or decreased levels, respectively, for the indicated parameters.

isotype-specific fine-tuning of spatiotemporal dynamics of the microtubule network to support critical cell functions.

## Materials and Methods

The human non–small cell lung cancer cell line NCI-H460 was cultured and validated as described previously (Parker et al, 2016). Control nonsilencing shRNA and βIII-tubulin shRNA-expressing NCI-H460 clones were cultured and validated as described previously (Mccarroll et al, 2010).

### Gene editing of NCI-H460 cells

The βIII-tubulin gene (*TUBB3* gene sequence; NCBI gene ID: 10381) is a large 14-kb gene, which has highly homologous exonic sequences compared with other tubulin genes and poorly defined regulatory regions. To selectively target the βIII-tubulin gene, two zinc-finger nucleases custom designed to specifically target the 5′ end of intron 1 of the *TUBB3* gene (Sigma Aldrich) were encoded within pZFN vectors under control of a CMV promoter and BGH polyA tail (Sigma Aldrich). Donor cassettes were designed to encode the coding sequence of the β-tubulin protein with modified C-terminal tail region. Because of the poorly defined nature of the *TUBB3* regulatory elements (Dennis et al, 2002), this coding sequence was driven by a CMV promoter. To identify cells in which homologous recombination was successful and to track the tubulin proteins in live cells, the tubulin coding sequence was fused to a GFP C-terminal tag separated by a 14–amino acid-long flexible linker region that has been extensively validated in microtubule dynamics studies to not perturb microtubule dynamics (Straight et al, 1997; Ludin & Matus, 1998; Heidemann et al, 1999; Rusan et al, 2001; Gan et al, 2010) and verified by in silico structural prediction modeling to have minimal influence on the structure and disorder propensity of the tubulin and GFP proteins. These elements were flanked by homology arms surrounding the ZFN target site (*TUBB3* gene sequence; NCBI gene ID: 10381; Sigma Aldrich) and were subcloned into the pTracerEF-V5His plasmid (Invitrogen) to generate the pDNR vector (Sigma Aldrich). The βIII-tubulin (NM_006086.3) and βI-tubulin (NM_178014) body sequences were amplified from existing *TUBB3*-pTracer and *TUBB*-pd2EGFP-N1 (Gan et al, 2010) vectors, respectively, using the HotStar HiFidelity PCR kit with QC solution (Qiagen). Modification of the β-tubulin C-terminal tail regions was achieved by using modified reverse primer sequences. The primer sequences used were full-length βIII-tubulin forward GCGGAATT-CACCATGAGGGAGATCGTGCACATCC, reverse ATTGCGGCCGCTCTTGGG-GCCCTGGGCCTCC; full-length βI-tubulin forward GCGGAATTCACCATG-AGGGAAATCGTGCACATCC, reverse ATTGCGGCCGCTGGCCTCCTCTTCGG-CCTCCTCACCGAAATCCTCCTCTTCCTCGGCCGTGGCGTC; truncated βIII-tubulin forward GCGGAATTCACCATGAGGGAGATCGTGCACATCC, reverse ATTGCGG-CCGCTCGTGGCGTCCTGGTACTG; and βIII-tubulin body with βI-tubulin tail forward GCGGAATTCACCATGAGGGAGATCGTGCACATCC, reverse ATTGCGG-CCGCTGGCCTCCTCTTCGGCCTCCTCACCGAAATCCTCCTCTTCCTCGGCCGTG-GCGTC. Amplified PCR products were purified (QIAquick Gel Extraction Kit; Qiagen) and subcloned into the pDNR vectors by EcoRI/NotI digestion (Promega) and T4 ligation (Invitrogen). Plasmids were propagated in competent JM109 *Escherichia coli* and extracted using the Qiagen Plasmid Maxiprep kit (Qiagen) for all transfections. All plasmids were sequenced to confirm their correct insert sequences (Ramaciotti Centre, University of New South Wales [UNSW Sydney]).

NCI-H460 cells were simultaneously nucleofected (Nucleofector II; Lonza) with two pZFN and one pDNR plasmid (Solution T, Program T-020; Lonza). Nucleofected cells were then FACS sorted (FACSJazz; Becton Dickinson Biosciences) for GFP positivity. GFP-positive cells were maintained by single-cell cloning before being screened for knockout of the endogenous βIII-tubulin isotype expression (TUJ1 antibody) and expression of the higher molecular weight–modified β-tubulin protein with GFP tag by Western blotting (anti-βIII-tubulin antibody, clone TUJ1; anti-GFP polyclonal antibody; Cell Signaling Technology). Clones identified as having been correctly gene-edited were then confirmed and characterized for their correct gene editing at the protein level by Western blotting and immunofluorescence, at the transcript level by real-time PCR and by Sanger sequencing of mRNA, and at the DNA level by genomic DNA sequencing across the 3′ end of the insertion site at the *TUBB3* locus (Ramaciotti Centre, UNSW Sydney, and Garvan Molecular Genetics Facility).

### Western blotting

Western blotting was performed as previously described (Parker et al, 2016) using primary antibodies against α-tubulin (DM1A; Sigma Aldrich), GFP (Polyclonal 2555; Cell Signaling Technology), βI-tubulin (Clone SAP4GS; Abcam), βII-tubulin (Clone 7B9; Covance), βIV-tubulin (Clone ONS1A6; Abcam), total β-tubulin (Clone TUB2.1; Sigma Aldrich), acetylated tubulin (Clone 6-11B-1; Sigma Aldrich), and tyrosinated tubulin (Clone TUB1A2; Sigma Aldrich). The comparison of protein expression in gene-edited cells with the parental cell line was performed with a custom antibody against S55 of βIII-tubulin, which was validated in house for its specificity against this residue of the βIII-tubulin isotype (data not shown). GAPDH (Clone 6C5; Abcam) was used as a loading control unless otherwise specified.

### BrdU proliferation assay

The cellular proliferation rate was determined using the BrdU ELISA proliferation assay (Roche Life Science) as described previously (Parker et al, 2016).

### Immunofluorescence staining of the microtubule network

Cells cultured on poly-D-lysine–coated chamber slides (Thermo Fisher Scientific) were fixed in 4% paraformaldehyde/PBS, permeabilized with 0.1% Triton X-100/PBS, washed briefly, and then blocked in 10% FCS/PBS at room temperature followed by blotting with mouse α-tubulin antibody (Clone DM1A; Sigma Aldrich) diluted in 5% FCS/PBS overnight. The cells were incubated with Alexa-fluorophore–conjugated secondary antibodies (Invitrogen) before mounting in DAPI-containing mounting media (Vectashield). Slides were then imaged using a Zeiss LSM 880 microscope at 63× objective using optimal Nyquist sampling conditions and postprocessed in Zen (Zeiss) software to combine the 32-detector (Airyscan) image into a single image with improved signal and spatial resolution. For live-cell imaging of mitotic (spindle) microtubules, cells plated onto

poly-ᴅ-lysine–coated fluorodishes (World Precision Instruments) were imaged using a Leica TCS SP5 WLL confocal microscope using a 100× objective (NA 1.4), 488-nm laser, and GFP expression was detected by HyD detectors.

### Tubulin polymerization assay

Soluble and polymerized tubulins were separated as described (Kavallaris et al, 2001) and the fractions were then processed by Western blotting as described. Western blot membranes were probed for α-tubulin (Clone DM1A; Sigma Aldrich) and GFP (Cell Signaling Technology), visualized by X-Ray, and analyzed by densitometry (ImageJ).

### Microtubule assembly by EB3 tracking

Gene-edited NCI-H460 cells were nucleofected with 4 μg of mCherry-EB3-encoding pET28A plasmid as described previously (Montenegro Gouveia et al, 2010) (Solution T, Program T-020) before being plated onto poly-ᴅ-lysine–coated fluorodishes (World Precision Instruments) and incubated for 36 h. Time-lapse fluorescent images of EB3-mCherry motion were acquired at 37°C, 5% $CO_2$, in normal growth media using a Zeiss Spinning Disk Microscope with Yokogawa Spinning Disk, 561-nm laser and 63× NA 1.4 objective over 100 frames at a frame rate of 200 ms and recorded on an EM CCD camera (Quant EM). For STICCS analysis, cells were imaged using the Zeiss LSM 880 at 63× objective in fast Airyscan mode for 100 frames at a 200-ms frame rate using optimal Nyquist sampling conditions and postprocessed in Zen (Zeiss) software to combine the 32 detectors (Airyscan) image into a single time series with improved signal and spatial resolution.

For particle tracking analysis, the time-lapse images were preprocessed (photobleaching correction, background subtraction, and denoised), and each cell was defined by a region of interest using ImageJ. EB3 movement was then analyzed using the TrackMate plugin (Tinevez et al, 2017) with a LoG detector, and LAP tracker and quality and track displacement filters using optimized max-linking and gap-closing distances over each cell. The EB3 speed for all microtubule tracks in each cell was averaged to define the average microtubule growth speed per cell. At least 50 cells were analyzed per clone per condition in each of three independent experiments. To examine the effect of paclitaxel on cell behavior, cells were treated with 6 or 20 nM paclitaxel in normal growth media for 2 h immediately before imaging.

### MCAK-mediated microtubule depolymerization assay

Gene-edited NCI-H460 cells were nucleofected with an mCherry-MCAK–encoding plasmid (Ovechkina et al, 2002) before being plated on poly-ᴅ-lysine–coated chamber slides. 8 h after nucleofection, the cells were fixed and processed for α-tubulin immunofluorescence with AlexaFluor635 secondary antibodies using the immunofluorescence procedure described above. Chamber slides were imaged on a Leica TCS SP5 multiphoton confocal microscope using a 63× objective (NA 1.4) and 488-, 568-, and 635-nm lasers, multiphoton lasers, and PMT and NDD detectors as Z-stacks at 1-μm increments. Z-stack images were projected and regions of interest (ROI) were defined around cells that had been transfected with MCAK with a similar level of MCAK expression. ROI were also defined around

neighboring cells that lacked MCAK expression to serve as reference cells. The integrated density of α-tubulin was measured for each cell (ImageJ) and normalized according to the mean fluorescence intensity of MCAK compared with control cells expressing the full-length βIII-tubulin protein.

The same procedure was followed for analysis of MCAK activity in CHO cells; however, cells were incubated for 9 h after conucleofection with an mCherry-MCAK–encoding plasmid and the pDNR vectors before being fixed and stained for α-tubulin as described (Ovechkina et al, 2002). Projections of the Z-stack images acquired by microscopy were analyzed using CellProfiler (Carpenter et al, 2006) to calculate the mean fluorescence intensity of the cytoplasm for cells cotransfected with pDNR and mCherry-MCAK–expressing vectors, which were then normalized to the mean fluorescence intensity of nontransfected cells.

### STICCS

Single- and two-color STICCS was performed as previously described (Hebert et al, 2005; Toplak et al, 2012). ROI were defined around each cell and a Fourier immobile filter was applied in time to each pixel stack in the entire image series to remove the lowest frequency components (immobile objects). Each image was then divided into 16 × 16 pixel ROI and shifted 4 pixels in the x and y directions to map the entire field of view with oversampling in space. The time series was divided into overlapping 30 frame-sized time subsets of interest (TOIs) and shifted by 15 frames in time to cover the entire image series with oversampling in time. Spatiotemporal correlation functions were calculated for each ROI/TOI to measure vectors of the flow from the translation of the correlation peak as described (Brown et al, 2006). Noise vectors were eliminated using a vector similarity criterion for adjacent vectors, and all remaining vectors were plotted on the corresponding frames of immobile filtered image series.

The microtubule speed was calculated from the per-cell average speed of the autocorrelation function for the GFP channel. Similarly, the microtubule growth speed was calculated from the per-cell average speed of the EB3 autocorrelation function. The number of microtubule growth events was calculated as the per-cell average number of autocorrelated vectors in the EB3 channel. The cross-correlated speed was calculated as the per-cell average speed of the cross-correlation function. The proportion of cross-correlated events was calculated as the number of cross-correlated vectors as a percentage of the total number of autocorrelated vectors in the EB3 channel.

### Statistics

Data are presented as the mean ± SEM unless otherwise stated. Data were analyzed using ANOVA, two-sided t test, or nonparametric tests where appropriate (GraphPad Prism 5; Graphpad Software Inc.). A P-value <0.05 was considered statistically significant.

## Supplementary Information

# Acknowledgements

This work was supported by the Children's Cancer Institute, which is affiliated with the University of New South Wales (UNSW Sydney) and the Sydney Children's Hospital Network, and by grants from the Australian Research Council (ARC Discovery Grant DP140103290 and Australian Research Council Centre of Excellence in Convergent Bio-Nano Science and Technology CE140100036 to M Kavallaris), NHMRC (NHMRC Principal Research Fellowship APP1119152 to M Kavallaris), Cancer Institute NSW (CINSW Career Development Fellowship to JA McCarroll), UNSW Sydney (UNSW Research Excellence Award to AL Parker), Children's Cancer Institute (Children's Cancer Institute PhD Excellence Award to AL Parker), and Steggles PhD Scholarship (to AL Parker). This research used facilities at the Biomedical Imaging Facility, Mark Wainwright Analytical Centre at UNSW.

## Author Contributions

AL Parker: conceptualization, investigation, methodology, and writing—original draft.
WS Teo: investigation, methodology, and writing—review and editing.
E Pandzic: data curation and methodology.
JJ Vicente: data curation, investigation, methodology, and writing—review and editing.
JA McCarroll: supervision and writing—review and editing.
L Wordeman: conceptualization, funding acquisition, investigation, methodology, and writing—review and editing.
M Kavallaris: conceptualization, supervision, funding acquisition, methodology, and writing—review and editing.

## Conflict of Interest Statement

The authors declare that they have no conflict of interest.

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
