## [Reviewer comments · Life Science Alliance]

β -tubulin carboxy-terminal tails have isotype-specific effects on microtubule dynamics in human gene-edited cells

Amelia L. Parker, Wee Siang Teo, Elvis Pandzic, Juan Jesus Vicente, Joshua A. McCarroll, Linda Wordeman, Maria Kavallaris
DOI: 10.26508/lsa.201800059

Review timeline:	Submission Date:	25 March 2018
	Revision Received:	25 March 2018
	Editorial Decision:	26 March 2018
	Accepted:	28 March 2018

Report:

(Note: Letters and reports are not edited. The original formatting of letters and referee reports may not be reflected in this compilation.)

Please note that the manuscript was previously reviewed at another journal and the reports were taken into account in the decision-making process at Life Science Alliance. Since the original reviews are not subject to Life Science Alliance's transparent review process policy, the reports and author response cannot be published.

Please note that the manuscript was previously reviewed at another journal and the reports were taken into account in inviting a revision for publication at *Life Science Alliance* prior to submission to *Life Science Alliance*.

1st Editorial Decision	26 March 2018
---------------

Thank you for submitting your revised manuscript entitled " β -tubulin carboxy-terminal tails have isotype-specific roles in microtubule dynamics". Your manuscript was reviewed at another journal before, and the referee reports of this previous round of review were confidentially transferred to us with your permission.

The reviewers who evaluated your work elsewhere appreciated the use of cutting-edge and superior tools to study the question of isotype specificity in microtubule dynamics and the relationship between human beta-tubulin structure and function, focusing on the C-terminal tail. The reviewers noted that prior work in *C. elegans* also attributes isotype-specific functions of beta-tubulin to its C-terminal tail, and that C-terminal tails in general are known to influence dynamics of purified tubulin.

We appreciate your work and the point-by-point response and introduced changes you provided in response to the issues noted by the reviewers. We would therefore be happy to publish your paper in Life Science Alliance pending final revisions necessary to meet our formatting guidelines. I list below a few items you should pay attention to allow production of your manuscript.

Congratulations on this very nice work!